# Identifying the Molecular Fingerprint of Beta-Lactams via Raman/SERS Spectroscopy Using Unconventional Nanoparticles for Antimicrobial Stewardship

**DOI:** 10.3390/antibiotics13121157

**Published:** 2024-12-02

**Authors:** Vinicius Pereira Anjos, Caroline Guimarães Pançardes da Silva Marangoni, Rafael Nadas, Thiago Neves Machado, Damaris Krul, Luiza Souza Rodrigues, Libera Maria Dalla-Costa, Wido Herwig Schreiner, Denise Maria Zezell, Arandi Ginane Bezerra, Rafael Eleodoro de Góes

**Affiliations:** 1Laboratório de Biofotônica, Centro de Lasers e Aplicações, Instituto de Pesquisas Energéticas e Nucleares, Universidade de São Paulo, São Paulo 05508-000, Brazil; viniciusanjo1@usp.br (V.P.A.); zezell@usp.br (D.M.Z.); 2Laboratório Fotonanobio, Programa de Pós-Graduação em Física e Astronomia, Universidade Tecnológica Federal do Paraná, Curitiba 82590-300, Brazil; carolinemarangoni.2020@alunos.utfpr.edu.br (C.G.P.d.S.M.); thiagomachado@alunos.utfpr.edu.br (T.N.M.); wido@fisica.ufpr.br (W.H.S.);; 3FabNS, Parque Tecnológico de Belo Horizonte-BHTec, Universidade Federal de Minas Gerais, Belo Horizonte 31310-260, Brazil; nadas@fabns.com (R.N.); 4Instituto de Pesquisas Pelé Pequeno Príncipe, Faculdades Pequeno Príncipe, Curitiba 80250-060, Brazil; damaris.krul@aluno.fpp.edu.br (D.K.); luiza.rodrigues@pelepequenoprincipe.org.br (L.S.R.); libera.costa@professor.fpp.edu.br (L.M.D.-C.)

**Keywords:** beta-lactam, drug monitoring, antibiotics, Raman, SERS, nanoparticles, biophotonics

## Abstract

**Background/Objectives:** Beta-lactam antibiotics, derived from penicillin, are the most used class of antimicrobials used for treating bacterial infections. Over the years, microorganisms have developed resistance mechanisms capable of preventing the effect of these drugs. This condition has been a significant public health concern for the 21st century, especially after predictions that antimicrobial resistance could lead to 10 million deaths annually by 2050. The challenge of developing new antimicrobials brings with it the need to ensure the efficacy of existing ones, hence the importance of developing fast and low-cost monitoring techniques. **Methods:** In this study, we present an alternative based on nanophotonics using Surface-Enhanced Raman Spectroscopy (SERS) mediated by nanoparticles for the detection of antimicrobials, with emphasis on some beta-lactam antibiotics commonly prescribed in cases of critically ill patients. It is a sensitive and accurate technique for drug monitoring, allowing for rapid and specific detection of its molecular signatures. This approach is crucial to address the challenge of antimicrobial resistance and ensure the therapeutic efficacy of existing treatments. **Results:** Our experiments demonstrate the possibility of identifying spectra with characteristic vibrations (fingerprints) of these antimicrobials via SERS. **Conclusions:** Our results point to new strategies for molecular monitoring of drugs by optical techniques using unconventional nanoparticles.

## 1. Introduction

Since the discovery of penicillin in 1928, considered one of the most critical advances in the history of medicine [1], many infectious diseases are now treated with antimicrobials, thus reducing morbidity and mortality [2]. Over time, microorganisms, causative agents of contagious diseases, have been creating resistance mechanisms, mainly due to the indiscriminate use of antimicrobials, thus reducing the efficacy of these drugs [3,4].

In 2014, research conducted by O’Neill estimated that by the year 2050, there could be up to 10 million deaths/year due to the absence of efficacious treatment to combat Multidrug-Resistant Bacteria (MDR) [5,6]; This prediction remains a great truth, as recent studies have shown that 4.95 million deaths in 2019 alone from Antimicrobial Resistance (AMR) happened [7,8]. AMR is the most significant global concern for health services in the 21st century, mainly because it implies an increase in the length of hospital stay and requires the use of broad-spectrum antimicrobials in patients affected by infections, increasing the costs for clinical treatments [2,3,5].

The exacerbated use of antibiotics (ATBs) has led to the increase of resistant strains and, consequently, harming complex procedures in the health area, which has the intervention of ATBs in a necessary way, harming these procedures and increasing the chance of death of patients [9]. An example of this inappropriate use was the COVID-19 pandemic, which brought even more concerns about AMR because many patients who had mild or moderate symptoms due to a viral infection took ATBs as a therapeutic measure [3,10,11]. 

The current lack of new antimicrobials to replace those that become ineffective brings urgency to the need to protect the efficacy of existing drugs [12]. After O’Neill’s publication, the World Health Organization (WHO) has been working tirelessly to combat AMRs. In 2015, at the 68th World Health Assembly, the WHO launched fundamental objectives for this confrontation, and among them were optimizing the use of antimicrobials and developing sustainable approaches for the development of new medicines and diagnostic tools [13].

The Antimicrobial Stewardship program has been worked on since the 1990s; however, in recent years, with the incorporation of One Health (animal, plant, and environmental health), the term has become popular, bringing an even more multidisciplinary vision [14]. Given that, several studies from academia and the pharmaceutical industry have tried to identify methods to optimize the use of antimicrobials to combat resistant pathogens [2].

Some strategies to find a solution to combat AMR aim to improve understanding of the molecular processes involved in drug resistance mechanisms, origin, evolution, and distribution across bacterial and genomic populations [15]. Combating AMRs involves, among other approaches, monitoring ATBs.

Traditional methods such as chromatography and immunoassay require a longer usage time to obtain results and often require several steps for sample identification and quantification, suppressing valuable molecular information for drug monitoring [16]. In this context, studies in the field of nanotechnology have been promising in health basic and translational research, as the ability to manipulate, monitor, and identify structures at the molecular scale has significantly impacted the search for diagnostic methods and drug administration [17].

In line with these necessities, Raman spectroscopy has several applications in the health area, such as polymorphic study, identification of raw materials, identification of “counterfeiting”, and determination of the quantity and homogeneity of active pharmaceutical ingredients (API) [18,19,20]. This technique involves illuminating a material and detecting the scattered light, which carries distinct spectral information. Part of the incident energy interacts with the molecular vibrations of the material, resulting in characteristic Raman peaks that serve as a molecular fingerprint. These peaks enable the identification of chemical bonds and functional groups present in the sample, making Raman spectroscopy a valuable photonic tool for both qualitative and quantitative analysis. The set of these vibrations, which appear in the form of peaks in the spectrum of the scattered light, forms a fingerprint of the material [21,22]. Furthermore, its versatility allows non-destructive analysis of solid, liquid, and gaseous samples, with applications extending over fields such as materials science, environmental monitoring, pharmaceuticals, and biomedicine. In healthcare, Raman spectroscopy holds transformative potential, enabling non-invasive measurement of drug concentrations in a patient’s bloodstream, minimizing the need for extensive blood collection or other invasive procedures [16]. Additionally, it facilitates the identification of protein-based drug substances in a non-destructive manner [23,24], reinforcing its critical approach in advancing precision medicine and pharmaceutical quality control.

However, Raman spectroscopy has limitations related to fluorescence or low-intensity scattered signals when the analyte concentration is small. Surface-Enhanced Raman Spectroscopy (SERS) comes as a solution to these problems [18,19]. When a material (analyte) is adsorbed in the vicinity of nanostructures, interactions occur, such as charge transfer (a chemical effect) and also a significant increase in the electric field of the incident light (a physical effect called plasmonic resonance, more accentuated in the presence of metallic nanostructures) [16,18,25,26]. The chemical effect mechanism involves the interaction of adsorbed molecules with the metallic surface (chemisorption), resulting in polarizability changes or metal-adsorbate charge transfer, leading to the creation of new molecular states due to this direct metal-analyte interaction and peak shifts in SERS spectra as compared to Raman. On the other hand, the plasmonic electric field enhancement allows for the occurrence of SERS even in circumstances in which analytes are not directly adsorbed on the metal but are within a few nanometers from it (in places called hotspots). Currently, most authors agree that the chemical mechanism provides enhancements of only a few orders of magnitude, whereas the plasmonic enhancement mechanism is the dominant contribution to SERS [16], and can imply amplification of the Raman signal by factors of up to 10^12^, thus enabling even the detection of a single molecule [27,28]. In this sense, exploring specific interactions between different combinations of analytes and nanomaterials (nanoparticles and nanostructured surfaces) is an open field of investigation for the determination of sensing capabilities in diversified configurations.

The usual laboratory methods for drug monitoring were compared with the SERS technique in Table 1. This comparison highlights that with further development and refinement for each specific sample, it may become a powerful tool for drug monitoring in the future. This approach also depends on parameters such as laser power, exposure time, type of acquisition point, map, or image [29], which may become a standard tool for drug monitoring in the future.

The potential of SERS spectroscopy is, therefore, an auspicious starting point in science and technology, with evident implications in the field of nanomedicine and the area of health, including for drug monitoring, given the potential for speed, sensitivity, and specificity of the Raman technique [10,16,30].

This work presents the SERS-collected spectra of beta-lactams to register their active principle molecular structure. These data are necessary to identify any changes in their structures that may indicate bacterial resistance, rendering their mechanism of action ineffective. This technique has the potential to be used as a tool for monitoring antimicrobials, with emphasis on beta-lactams, an antimicrobial widely used in the treatment of patients with bacterial infections. 

### Beta-Lactam Antimicrobials

Beta-lactams have their origin in penicillin, and even after so many years of their discovery and clinical application, they continue to be the most prescribed and most important class for the treatment of bacterial infections, as they are safe and have low toxicity [31,32,33]. They have mechanisms of action in the synthesis of the bacterial cell wall, acting specifically on the cross-linking of peptidoglycans, causing lysis/death of the bacterium [34].

Beta-lactam antimicrobials (such as penicillins, cephalosporins, or carbapenems) are similar in the molecular formation (D-Ala-D-Ala) that constitutes the cell wall of the peptidoglycan, in addition to having highly reactive carbonyl in the beta-lactam system, ensuring the acetylation of the serine residue in the active center of the transpeptidases involved in the final steps of peptidoglycan synthesis [10,34]. However, what makes them a large class is that they comprise the same structural nucleus of 3 carbon atoms and 1 nitrogen atom, which forms the beta-lactam ring, which is the bactericidal active ingredient of the drug [33].

The beta-lactam ring must be attached to at least one radical in the molecular structure to confer its pharmacological activity. In this way, the radicals differentiate the types of beta-lactams from their structure, giving them chemical changes capable of generating characteristics such as affinity to the receptor, spectrum of action, and even different forms of resistance [32].

## 2. Materials and Methods

### 2.1. Selection of Antimicrobials

In this study, three representatives of antimicrobials were selected (Figure 1) due to their chemical similarity. Representing the penicillin subclass, we used ampicillin, a drug widely monitored by Raman/SERS research, which can be compared with the literature for data analysis. Meropenem was selected to represent the carbapenems due to its wide use in the clinical area. Finally, representing cephalosporins, ceftazidime was the choice for being a drug little explored in monitoring based on Raman/SERS spectroscopy.

### 2.2. Sample Preparation

The ATBs were purchased in pure formulations from the manufacturer Sigma Aldrich. The samples used were in powder form, and at first, these solids were prepared for the experiments to obtain the corresponding Raman spectra. Ampicillin (code A0166, 371.39 g/mol), ceftazidime (code C3809-1G, 549.58 g/mol), and meropenem (code Y001252, 437.51 g/mol) were studied. Because ATBs for clinical use have adjuvants to their chemical formula, this pure chemical formulation was chosen to guarantee the corresponding reference spectra.

The samples were prepared using borosilicate microscopy glass slides (Knittelglass, Bielefeld, Germany) as substrates. In the Raman experiments, pure solid samples in powder form were used for measuring the Raman spectra of each selected ATB. Powder samples were also diluted in water (deionized, conductivity less than 5 ms) at concentrations of 100 mm; thus, a 1 μL drop was deposited on the glass slide and left to dry. These dried drops were used as a reference since, for each ATB of choice, a clear Raman signal could be obtained and compared with the corresponding powder Raman spectra.

The ATB solutions were then further diluted in pure water several times, and the dried drops were tested over the slides without nanoparticles until Raman signals could not be detected (under the same experimental configurations used for the 100 mM concentration drops). Once no Raman signal could be measured, we used those over-diluted samples as a starting sample concentration to verify SERS occurrence. For the SERS substrate preparation, a 1 μL drop of each nanoparticle colloid (concentration of 0.01 mg/mL) was deposited on the glass slide and left to dry for film formation, therefore constituting our SERS substrates upon which ATB drops were deposited. In the particular case of CoNP, which is a paramagnetic material, the slide was placed over a neodymium magnet (ca. 270 mT) to obtain a more homogeneous substrate. Therefore, for the SERS experiments, ATB concentrations that depended upon the specific nanoparticle-ATB combination were used.

### 2.3. Raman Spectra Measurement

An amount of 1 mg of each ATB powder was deposited on a glass slide. Next, the slide was positioned on the spectrometer, directing the laser toward the target to acquire the Raman spectra of the ATBs in their pure form. The spectrometer used was Anton Paar’s Cora 5001 model, with a resolution of 6 cm^−1^ to 9 cm^−1^ for 785 nm and a resolution of 12 cm^−1^ to 17 cm^−1^ for 1064 nm. The equipment operated at 785 nm and 1064 nm wavelengths with power ranging from 2 µW to 450 mW. The measurements were conducted with acquisition times between 1 and 10 s, and different numbers of spectra accumulations were used for the various molecules analyzed. 

The Witec alpha 300R confocal Raman microscope (Oxford Instruments, Abingdon, UK) was used to obtain the spectra in visible light. The equipment has a stage to position the sample, providing a lateral resolution of 200 nm and a vertical resolution of 500 nm. Lasers of 532 nm and 633 nm were used, along with two spectrometers with a resolution of 0.02 cm^−1^. The acquisition time was adjusted in the 0.1 to 10 s range, and different accumulation numbers were used. 

Regarding spectral data processing, OriginLab software (version 2024b) was used for analysis, using mathematical methods to obtain “clean spectra”, in which the vibrational modes of the spectral signature are highlighted. The “raw” spectra clearly evidence a change in the baseline and interference caused by cosmic rays. Pre-processing of the raw dataset involved baseline calibration using the asymmetric least squares smoothing method and eliminating cosmic rays using the abnormal peak detection method. Noise removal was performed by the application of smoothing using the Savitzky–Golay convolution method (points of window: 25, polynomial order: 2).

### 2.4. Nanoparticle Synthesis and Characterization

The LASiS (Laser Ablation Synthesis in Solution) technique is based on the principle that when a laser beam irradiates a solid-state material in a liquid medium, the energy provided by the laser pulse can be absorbed by the target and lead to the formation of an expanding plasma cloud containing the ablated material. This is accompanied by the emission of a shock wave that releases energy into the surrounding liquid. When the plasma cools, it releases its heat into the liquid, which turns into hot vapor. This leads to an oscillating cavitation bubble containing both the ablated matter and the liquid vapor, where particles form [35,36].

The formation of nanoparticles can occur through various mechanisms and is influenced by multiple laser parameters, such as pulse duration, wavelength, energy, repetition rate, and the materials involved. When ultrashort laser pulse generators are used, the ablation rate is so fast that the solvent does not heat up when struck by the laser, resulting in the absence of heat transfer. This keeps the solvent cool, allowing laser ablation in solvents with a low boiling point [37].

The choice of solvent composition, target morphology, laser fluence, pulse duration, and radiation time all play crucial roles in the control and repeatability of nanoparticle synthesis [35]. For this work, the Au targets, which are the most commonly used in SERS spectroscopy, were selected, and we used unconventional metals that also have plasmonic effect results in this research: Co, Cu, Bi, and V. The metal target foils were purchased from Sigma-Aldrich with 99.99% purity.

As emphasized by Bezerra et al. [38], most plasmonic research on SERS is generally conducted by using the noble metals Au and Ag [39,40]. However, recently, intense research has been carried out demonstrating that other metals can show significant improvements in the electric field, comparable to noble metals; for example, Al, Ga, In, Sn, Tl, Pb, and Bi have been investigated for localized surface plasmonic resonances (LSPR) and SERS, demonstrating amplification factors comparable to those of gold nanoparticles. 

To standardize the synthesis of the colloids in this research, we tried to maintain a pure water column of 2 cm above the metallic target. Thus, the ablation time would remain proportional for each target. Those targets that had a quantity of water of 8 mL were irradiated for 3 min. The nanoparticles were synthesized by the laser ablation technique in solution with the Brilliant B model equipment, operating at wavelengths of 1064 nm and 532 nm, Q-switched at 338 μs, delivering pulses of 5 ns with a total energy of 220 mJ (200 mJ for IR and 20 mJ for green) and a fluence of 2.83 × 10^7^ J/m^2^ in a beam waist of 30 μm.

At the end of this process, colloidal solutions formed only of metallic nanoparticles suspended in water (without additives) were obtained, which were characterized for their optical absorption (UV-Vis spectrophotometry) and size dispersion through dynamic light scattering (DLS).

### 2.5. SERS Spectra Measurement

In places with a higher electric field concentration, the Raman scattering signal intensifies more significantly. As the coverage of NPs and the analyte tends to not be uniform on a slide, both for the map and for singular measurements, question marks were taken in an exploratory manner. SERS spectroscopy data were obtained from two different spectrometers. The first equipment will be the same as the one used to obtain the Raman spectrum, that is, the Witec Alpha 300R equipment, operating with an excitation laser at 532 nm, a 50× objective, and a power of 10 mW.

For the data processing of the SERS spectra, the same methodology applied previously in the graphs referring to the Raman signal is also used here. Measurements were also performed with the Porto Core system, with a resolution of 5 cm^−1^, operating at 633 nm (visible) with power between 2.0 μW and 20 mW, and a 60× oil objective (NA = 1.4). Different acquisition times and numbers of spectra accumulations were used to analyze the various molecules. In addition, other powers were used for the same sample to obtain the spectrum with the lowest possible power.

#### SERS Map

For the vanadium nanoparticles (VNP) with ampicillin combination, a hyperspectral image was obtained (map), where each pixel of the image corresponds to an associated spectrum of the sample [41]. The measured map was treated using the FabNS PortoFlow software (v1.17). The deposition on the glass is performed with the two droplets (solution and NP) dripped in different places but with a region of intersection, as shown in Figure 2. The spectrometer’s path was unidirectional, so no two measurements were taken at the same point.

A 5 μL of VNP drop was deposited on the glass slide and expected to dry. Due to its intense yellow color, the process of depositing a droplet with ATB diluted in 0.5 mL of pure water was facilitated (this sample is transparent), as described in the methodology in Figure 2. After drying, which took about 10 min, the glass slide was placed in the spectrometer and adjusted so that the laser made a map in a region of 50 μm × 50 μm (256 points). The time to obtain the data was approximately 7 h.

## 3. Results

Initially, experiments were conducted to obtain the Raman signals of the chosen antimicrobials, whose spectra are shown in Figure 3. It is possible to observe that the different antimicrobials show a similar pattern. There is a characteristic region of vibrations between 500 cm^−1^ and 2000 cm^−1^ that can be associated with the characteristics of the molecule of each ATB (the “fingerprint region”); in this region, there are vibrations such as the aromatic rings and the breathing modes, including stretching of the groups related to the beta-lactam ring. This result forms the basis for creating a library of spikes associated with each antimicrobial, enabling their identification, for example, in blood samples from patients to be analyzed in future studies. 

Figure 4 shows the Raman spectroscopy measurements between 250 and 2000 cm^−1^, made with four different wavelengths for ampicillin. The main vibrational bands are listed in Table 2, with their respective assignments based on comparisons with previously published data. 

This selected region has many bands and is rich in structural information. A very intense vibration is evident in the region of 1002 cm^−1^, which is associated with the benzene ring of the molecule. The last two vibration peaks are related to the vibrations of the beta-lactam ring. It should be noted that the measurements performed are remarkably close to those made (or predicted) in the theoretical frameworks.

It was also apparent in Figure 4 that the spectral resolution for the same spectrometer depends on the wavelength of the laser. The vibration-related peaks for the laser operating at 532 nm were more defined and distinguishable than those at 632 nm (Witec Raman spectrometer). The same is expected to occur in IR, although to a lesser degree (Anton Paar Raman spectrometer, Graz, Austria). An example of this is seen in the region of 1602 cm^−1^, which is better resolved by presenting a vibrational peak at 1584 cm^−1^, as well as a broadband band at 1456 cm^−1^, that resolves into two peaks measured at 1539 cm^−1^ and 1459 cm^−1^.

Figure 5 shows the measurements of Raman spectroscopy in the frequency range between 250 and 2000 cm^−1^, made with four different wavelengths for meropenem. The main vibrational bands are listed in Table 3, with their respective attributions based on comparisons with data already published in the literature.

A wide range of spectral information is evident in the corresponding range between 250 and 2000 cm^−1^. The vibrational modes corresponding to the vibrations of the C-O-H band in the carboxyl group are located in the region between 653 and 690 cm^−1^. The most intense vibration at 1556 cm^−1^ is associated with the beta-lactam and pyrrolidine rings. Other vibrations related to the beta-lactam ring are associated with vibrations at 1392, 1418, and 1754 cm^−1^. In this experiment, it is also possible to identify the resolving power when we observe the vibrational modes in the bands close to 1266 cm^−1^, which are better resolved at 532 nm with a second peak at 1284 cm^−1^. This phenomenon is also noticeable at 1388 cm^−1^, which shows a second peak at 1368 cm^−1^ in visible light.

Similarly, Figure 6 shows the measurements of Raman spectroscopy in the frequency range between 250 and 2000 cm^−1^, made with four different wavelengths, now for the ATB ceftazidime. The main vibrational bands are listed in Table 4, with their respective attributions based on comparisons with data already published in the literature.

Ceftazidime is an ATB that has yet to be explored in the literature since little research material has been found for its band assignment. However, the three most intense peaks were detected by [45], which shows a graph very similar to the one measured here. The authors found peaks in the region of 1028 cm^−1^ (measured in 1026), 1506 (measured in 1502), 1586 (the most intense—measured in 1582), and 1641 cm^−1^ (measured in 1646). In this article, the authors do not associate the vibrations with any part of the molecule, as this was not the objective of the work. According to [39], the most intense Raman peak near 1586 cm^−1^ is due to the C=O band of the elongation type, in which the measurement made was at 1582 cm^−1^, varying little about the measurement presented. An associated vibration is also given by [42] in the region of 1710 cm^−1^.

The nanoparticle colloids were prepared using the previously reported procedure where the target of each metal was immersed in pure water and irradiated by the laser. The optical characterization of the suspensions was performed by UV-Vis spectroscopy, and through DLS we obtained the size distribution. In Figure 7A, we present the optical extinction of all colloidal nanoparticles. Noticeably, these colloids exhibit plasmon bands that are characteristic of spherical nanoparticles; whilst gold and copper present peaks in the visible region (around 520 nm and 600 nm, respectively), the other elements present peaks in the ultraviolet region. Figure 7(B1–B5) summarizes DLS measurements for all nanoparticles. DLS typically provides larger diameter values compared to those obtained from more direct measurement techniques; the discrepancy arises because DLS measures the diffusion behavior of particles in solution, accounting not only for the particle’s core size but also for the surrounding solvation layer and any particle-particle interactions. Therefore, DLS tends to overestimate particle size, and the values for the average size and dispersion measured are biased to larger particles. Yet, this technique allows us to compare the size distribution of nanoparticles from the different materials used in our experiments. It is worth mentioning that the broader size distribution observed in Figure 7(B1–B5) arises from the intrinsic nature of the LASiS process, where rapid and localized energy deposition on the target material gives rise to particles of varying sizes within a single ablation event [35]. In addition, factors such as the laser parameters, pulse duration, and the dynamics of bubble formation and collapse in the liquid environment depend on the specific metal target used in the synthesis, which explains the heterogeneity in nanoparticle size observed for the different colloidal solutions in Figure 7(B1–B5).

An interesting feature in the experiments performed with ampicillin illustrates the importance of using nonconventional nanoparticles as SERS substrates as alternatives to noble metals. Although some experiments were performed with silver nanoparticles, it always led to local sample burning under the experimental conditions available for the SERS measurement setup. Similarly, when gold nanoparticles were used as substrates, no signal amplification could usually be observed since the integration time selected had to be noticeably short (to prevent sample burning). After several measurements, only a few SERS signals were detected with AuNP substrates, for which Figure 8 shows a typical spectrum.

The first relevant result measurement performed with AuNP is that even for a very short integration time, the characteristic spectrum of ampicillin was obtained, which leads us to conclude that this is indeed a SERS measurement. Another interesting fact is that the two peaks related to the beta-lactam ring (1692 and 1782 cm^−1^, on the lower right corner of Figure 8) were not evident in this measurement. Something that may justify this is that this measurement, unlike the other measurements reported here, was not obtained through a map that scanned an area of the sample but with exploratory point measurements in different regions of the sample. Perhaps if the integration time were longer, both peaks would be resolved; however, with a longer integration time, the sample would be burned locally. It is important to emphasize that biological samples and the drugs used in this research are fragile, so parameters such as excitation laser power and even NPs used can contribute to the sample being degraded quickly at the time of interrogation, which probably explains the large shifts observed in Figure 8. The most intense band was shifted by +94 cm^−1^, the 1201 cm^−1^ band was displaced by +164 cm^−1^, and the vibration at 1600 cm^−1^ was shifted to 1613 cm^−1^. In summary, despite their widespread use as SERS agents, gold nanoparticles did not perform satisfactorily and proved not to be efficient substrate materials in these experiments with ampicillin.

Ampicillin, therefore, was the ATB of choice to perform a proof of concept of the SERS effect with non-conventional metals because, in the literature, this ATB is recurrent due to its relevance and historical process of its synthesis and use [12,40,42,43]. Following the same procedure previously described, a very low concentration was prepared. This dilution now has experimental conditions imperceptible to Raman but ideal conditions for analysis of SERS spectra. The Raman spectrum corresponds to the signal of powder’s ATB, and under these same experimental conditions, after a 100-fold dilution, no Raman signal could be detected. Then, a thousand-fold dilution was performed to obtain the SERS spectrum, and the measurement was performed with the addition of colloid-containing BiNP. These spectra are shown in Figure 9.

It was observed that with a high power (5200 μW), the diluted sample still revealed its characteristic spectrum with the leading bands evidenced: 618, 1002, 1192, and 1608 cm^−1^. By decreasing the laser power approximately seven times (20.7 μW), the more intense band is noticeable (1002 cm^−1^) and with less intensity the vibration at 1608 cm^−1^. When the power of the pumping laser decreases further, the spectrum disappears. The laser intensity was decreased to 2 μW of power, and the vibration at 1002 cm^−1^ was amplified. Other bands are also noticeable, such as 912 and 1608 cm^−1^.

Another experiment with ampicillin, shown in Figure 10, is crucial to demonstrate the contribution of the Co colloid to the SERS effect. At the bottom, the characteristic Raman spectrum (powder sample—black line) is visible with high excitation power (5.2 mW). The diluted sample was then measured at a lower power (752 μW), and the characteristic peak at 1002 cm^−1^ is noticeable (red line). Once excitation power was further reduced, no signal could be detected (blue and green lines). After all, although the initial power decreased more than two orders of magnitude, it was still possible to verify the characteristic peaks of ampicillin once CoNP were added to the substrate. Therefore, another unconventional nanoparticle (Co) was used as an efficient alternative to the conventional ones (Ag and Au) in SERS experiments for ATB detection and characterization.

Several tests were performed with the ampicillin ATB. One of the methodologies used resulted in the spectrum obtained in Figure 11 using a Raman map.

The most intense ampicillin band (1002 cm^−1^ attributed to the benzene ring) was selected as reference. It was possible to find the characteristic spectrum in region A of the map, where the VNP was added to the ATB. In region B, where only the ATB was diluted, no signal is detected. The spectrum in Figure 11 consists of an arithmetic average taken over 15 spectra of different points selected in region A. It should be considered that the drying process of the sample causes the nanoparticles to be in greater quantity at the edge of the droplet (ring effect), hence the choice that this would be a good region for the measurement of the spectrum. In the Raman spectrum, the three intense peaks at the end of the graph are very characteristic, the first of which, as previously discussed, is also due to the benzene ring (1600 cm^−1^) and the other two to vibrations of the beta-lactam ring (1692 and 1782 cm^−1^). An exciting and relevant fact is that the measurement made with the diluted ATB did not show intensification of the two vibrations associated with the beta-lactam ring. However, they were subtly perceptible in the resulting spectrum. The peak at 1600 cm^−1^ underwent a displacement of +4 cm^−1^, while the peak associated with the thiazolidine ring underwent a displacement of +33 cm^−1^. These displacements can be compared with the values obtained already mentioned in Table 4. This measurement was not the first to be made, but it served as a parameter for interpreting the other results. Evidently, the nanoparticles contributed to the increase of the Raman signal since the integration time in the whole area was only 0.5 s and the power did not vary. In region (B), where the ampicillin fingerprint was not obtained, only a noisy signal was obtained; in region (A), with NP, the signal was found. 

For meropenem, Raman signals were measured exploratorily in different sample regions. Figure 12 illustrates those points where a measurable signal was obtained.

The purpose of this experiment was to show that the nanoparticles contributed to the signal intensification, and in the case where there was no NP, the characteristic spectrum was not obtained. Starting with the analysis with the Cu colloid sample, the laser power had to be raised to the equipment’s highest to yield a measurable spectrum. In comparison, the Co nanoparticle proved to be more efficient because even at lower power, the characteristic signal of the molecule was obtained. The integration time draws much attention, as a 150 times shorter time was used for cobalt colloids. It is suggested that CoNP was also more efficient in this regard, but further testing should be performed for more concrete statements. The most intense band, associated with the beta-lactam and pyrrolidine rings, at 1556 cm^−1^, underwent a displacement of −10 cm^−1^ for the Co colloid and −2 cm^−1^ for the Cu colloid. It is noticeable that the cobalt colloid intensified the signal more efficiently than the copper one because it presented more pronounced, narrower bands. Table 5 shows a summary of the obtained experimental results, listing the wavelengths for each ATB for which the Raman spectra were recorded and for which NPs an enhanced Raman spectra (SERS) could be obtained.

The process relies on the affinity between the sample and the colloid to obtain a SERS spectrum for any sample, which is ATBs at this rate, making each case dependent on the analyte and the NP (optimization of the method). Ceftazidime SERS spectra have already been acquired by [39] using AgNP. We suggest that further studies and simulations should be performed to assign all bands present in the powdered Raman spectrum of meropenem to specific molecular vibrations, for the present measurements only allow for concrete statements of the most intense peak, and more experimental tests using ceftazidime, e.g., for non-conventional NPs and band assignment.

## 4. Discussion

Our study showed that Au, Bi, Cu, Co, and V nanoparticles synthesized via laser ablation exhibit plasmonic properties, therefore presenting the potential for Raman enhancement in SERS experiments and SERS-based sensors. Although Au nanoparticles are constantly employed in SERS experiments, Bi, Cu, Co, and V characterize unconventional nanoparticle materials that also serve as mediators in plasmonic augmentation (hotspot creation), which is the basis for the SERS effect. In our experiments, the application of Raman/SERS spectroscopy revealed significant potential in monitoring beta-lactam ATBs of choice, named ampicillin, ceftazidime, and meropenem. Figure 3 illustrates the spectra of the three beta-lactam ATBs, in which regions between 1500 and 1800 cm^−1^ relate to C=O vibrations, an important beta-lactam fingerprint. By using statistical methods in conjunction with this technique, ATB action monitoring can be made possible in shorter time intervals. In addition, our approach does not require nanoparticle functionalization and can therefore be characterized as a label-free method, which is a novel and important field of SERS investigation [16,46,47,48]. This analytical approach highlights the advantage of its specificity and rapidity, which, together with the easy sample preparation process, could set the basis for new strategies for molecular monitoring of drugs. Whilst most studies using the SERS effect employ gold or silver nanoparticles, this work demonstrates the efficacy of unconventional metal nanoparticles, not only reducing costs but also acting as SERS agents in experimental setups where noble metal proved not to be efficient substrate materials, as shown in the experiments we performed with ampicillin. In fact, optical techniques using unconventional nanoparticles can benefit several scientific areas, especially in drug monitoring for health applications.

## Figures and Tables

**Figure 1 antibiotics-13-01157-f001:**
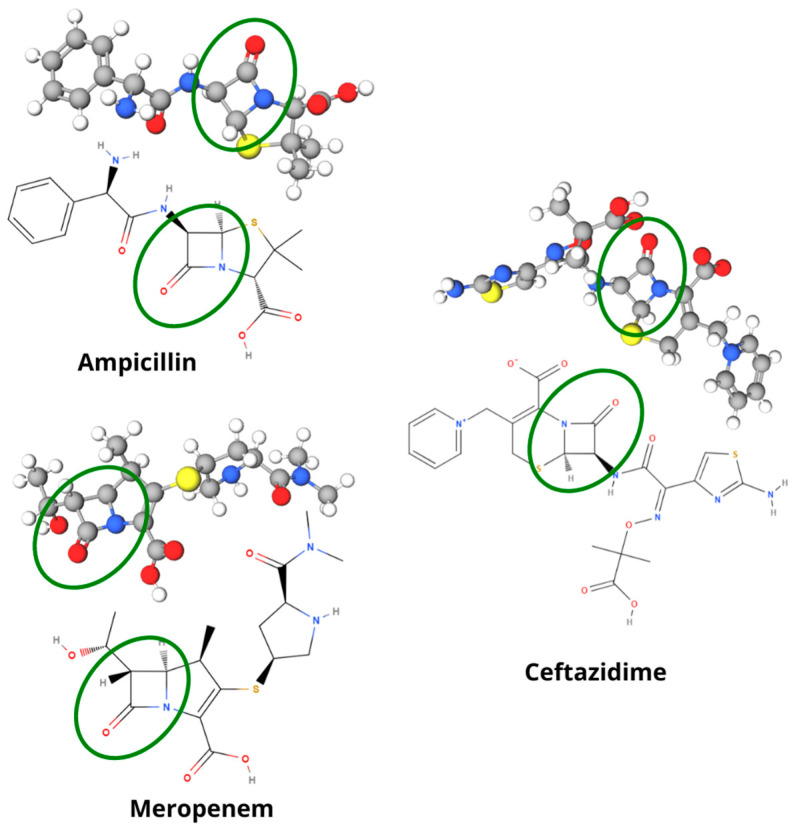
Molecular representation (2D and 3D) of the beta-lactam antibiotics studied: ampicillin, ceftazidime, and meropenem. Green circles highlight the beta-lactam rings.

**Figure 2 antibiotics-13-01157-f002:**
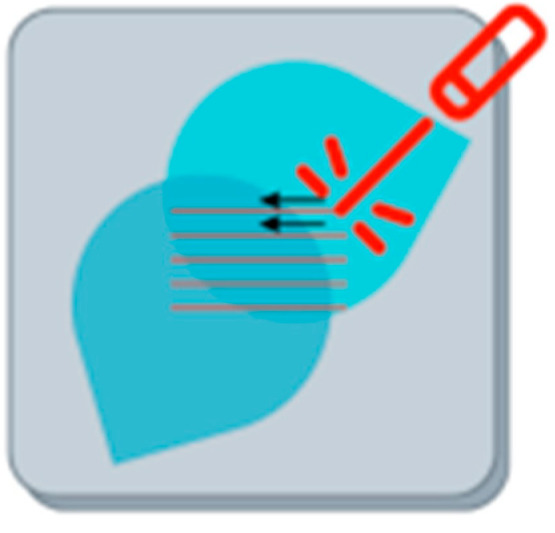
Diagram showing the sample deposition and the overlapping region of ATB (dark blue) and NPs (light blue) as well as the equipment scanning direction (indicated by the black arrows).

**Figure 3 antibiotics-13-01157-f003:**
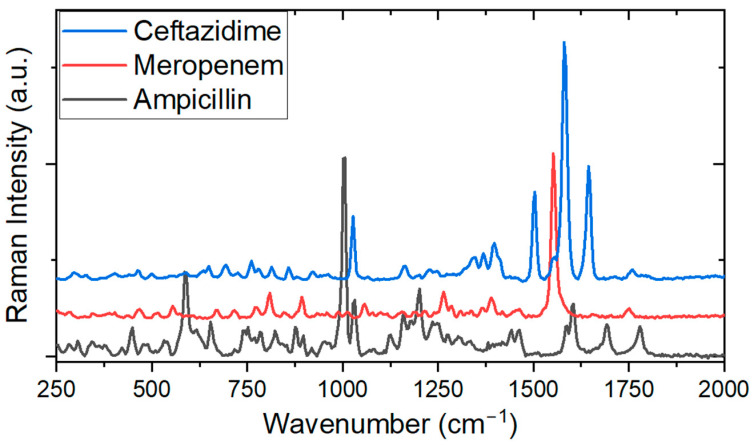
Raman spectra obtained for the different antibiotics. Each peak represents the detection of characteristic molecular vibrations.

**Figure 4 antibiotics-13-01157-f004:**
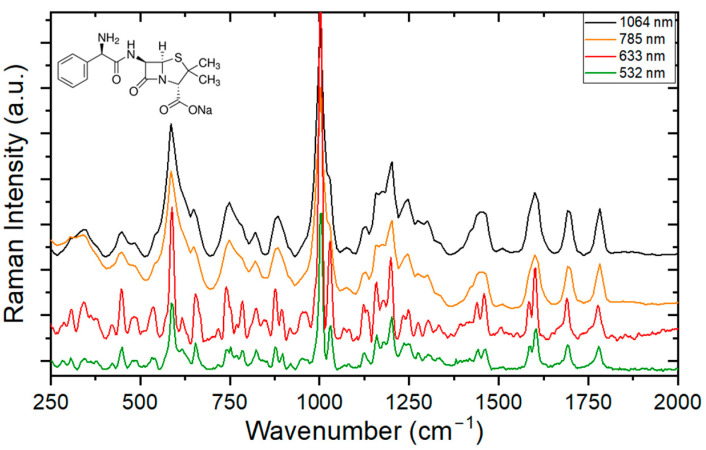
Raman spectrum of ampicillin (chemical structure on the top left) for different excitation wavelengths.

**Figure 5 antibiotics-13-01157-f005:**
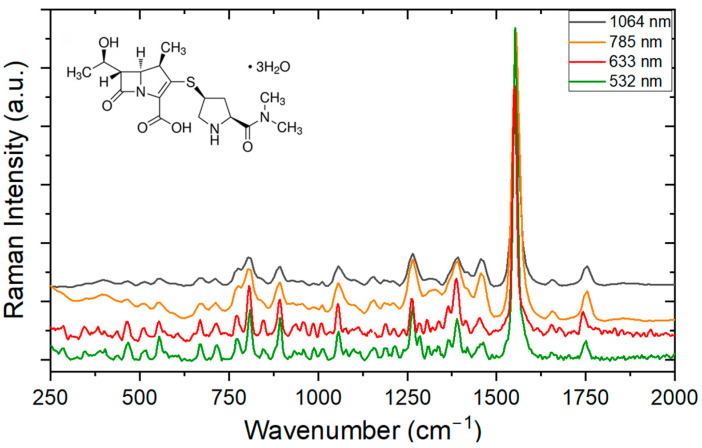
Raman spectrum of meropenem (chemical structure on the top left) for different excitation wavelengths.

**Figure 6 antibiotics-13-01157-f006:**
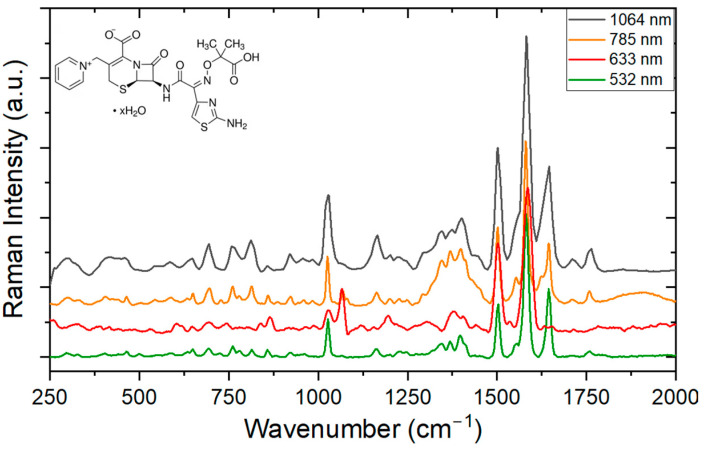
Raman spectrum of ceftazidime (chemical structure on the top left) for different excitation wavelengths.

**Figure 7 antibiotics-13-01157-f007:**
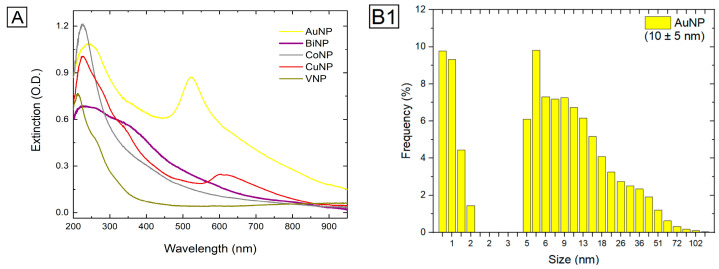
(**A**): UV-Vis of the synthesized nanoparticles. Size distribution and average sizes as measured through DLS for AuNP (**B1**), BiNP (**B2**), CoNP (**B3**), CoNP (**B4**), and VNP (**B5**).

**Figure 8 antibiotics-13-01157-f008:**
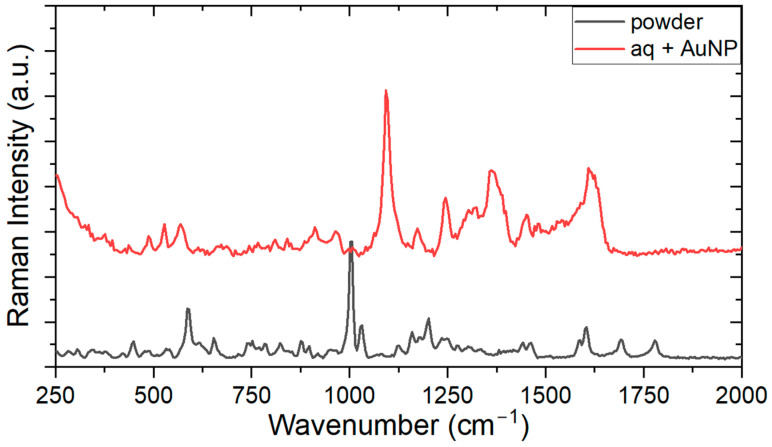
SERS measurement (10 mW laser, 0.1 s integration time) of ampicillin at 26.9 mM concentration and AuNP.

**Figure 9 antibiotics-13-01157-f009:**
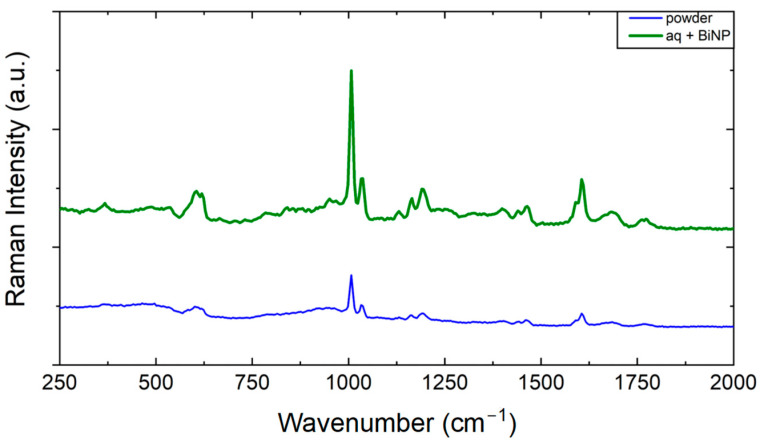
Raman spectrum (reference powder sample) and SERS with BiNP and ampicillin at a concentration of 0.1 mM.

**Figure 10 antibiotics-13-01157-f010:**
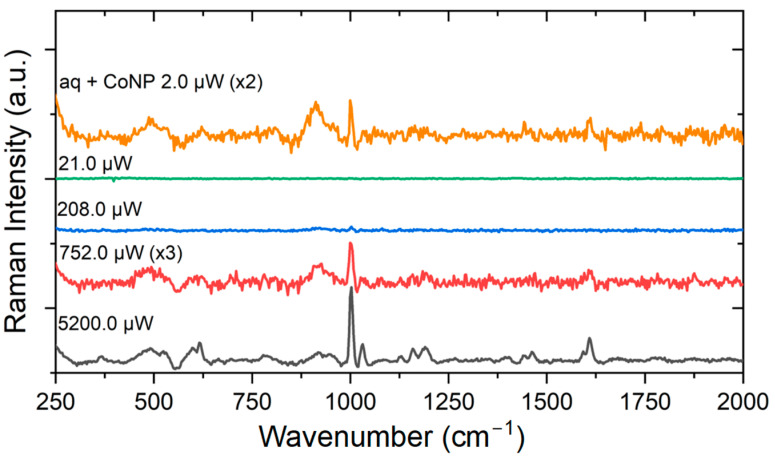
SERS measurement (laser with varying powers, integration time of 10 s) of ampicillin at a concentration of 26.9 mM and CoNP.

**Figure 11 antibiotics-13-01157-f011:**
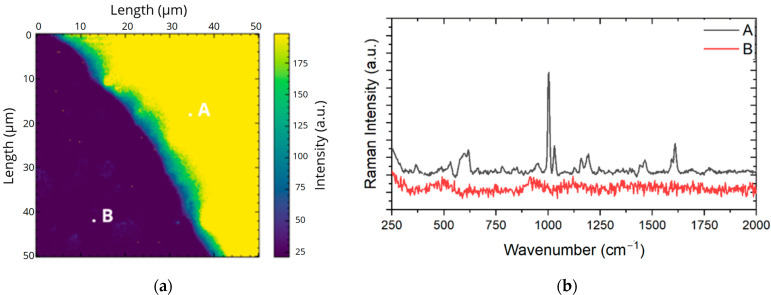
SERS measurement (laser at 5.2 mW, integration time 500 ms) of ampicillin at a concentration of 5.4 mM. (**a**) Map and (**b**) spectra for points A: ampicillin with VNP, and B: ampicillin only.

**Figure 12 antibiotics-13-01157-f012:**
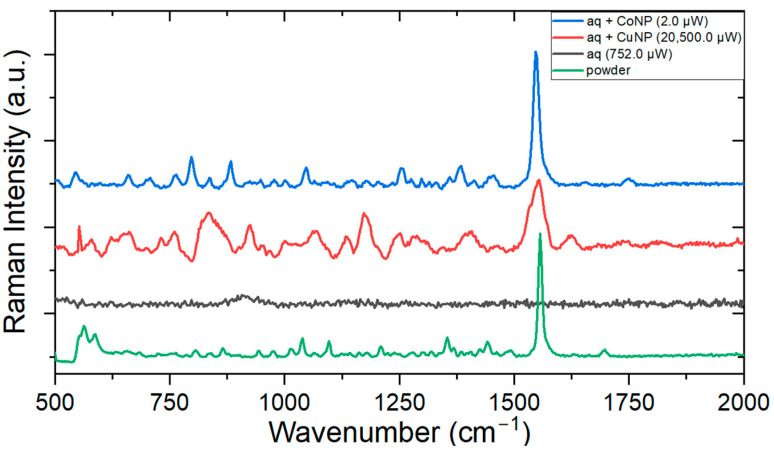
SERS measurement of meropenem at 4.6 mM concentration with CuNP (15 s integration time) and CoNP (0.1 s integration time).

**Table 1 antibiotics-13-01157-t001:** Comparison of analytical techniques used in drug monitoring with SERS.

Technique	Pros	Cons
Chromatography-based methods	Gold standard; robust methods with superior sensitivity; relatively interference-free; reduced drug/metabolite class cross-reactivity	Time-consuming; tests developed in the laboratory; interlaboratory variability; matrix effects; high technical knowledge required; high costs of installation, personal training, and method validation
Immunoassay platforms	Small sample quantity; run on automated, continuous, and random-access systems; there is no need for sample cleaning; multiplexing capabilities	Various steps to achieve analyte quantification, reduced specificity, and sensitivity. They often show significant bias; antibody cross-reactivity; interferences of bilirubin, hemoglobin, high lipid content, very high or shallow protein content, endogenous antibodies, various drugs, and metabolites
SERS-based methods	There is no need for sample preparation; fast measurement; multiplexing capabilities, availability of portable Raman spectrometers	Often high RSD (relative standard deviation) of SERS substrates; optimization of the method required for each drug

Adapted from [16].

**Table 2 antibiotics-13-01157-t002:** Ampicillin band assignment.

Wavenumber (cm^−1^) Measured	Wavenumber (cm^−1^) Theoretical ^1^	Assignment
586	583	Thiazolidine Ring
748	750	β-lactam ring
1002	1005	Benzene Ring (CC)
1600	1600	Benzene Ring (CC)
1692	1692	β-lactam ring (C=O)
1782	1782	β-lactam ring (C=O)

^1^ According to [12,40,42,43].

**Table 3 antibiotics-13-01157-t003:** Meropenem band assignment.

Wavenumber (cm^−1^) Measured	Wavenumber (cm^−1^) Theoretical ^1^	Assignment
712	715	Pyrrolidine ring
776	771	C-O-H
1184	1188	C-O-H
1266	1265	Dimethylcarbamoyl (C-N)
1392	1391	β-lactam ring (C-N)
1418	1417	β-lactam ring (C-N)
1556	1553	Pyrrolidine Ring/β-Lactam Ring (C=C)
1754	1751	β-lactam ring (C=O)

^1^ According to [44].

**Table 4 antibiotics-13-01157-t004:** Ceftazidime band assignment.

Wavenumber (cm^−1^) Measured	Wavenumber (cm^−1^) Theoretical ^1^	Assignment
694	689	C=S
762	749	β-lactam ring
812	850	C=C
920	930	C-O-C
1026	1025	CH
1502	1501	-
1582	1586	C=O
1646	1644	-
1764	1764	C=O

^1^ According to [39].

**Table 5 antibiotics-13-01157-t005:** Beta-lactam Raman/SERS interrogation.

Antibiotic	Raman at (nm)	SERS with
Ampicillin	532, 633, 785, 1064	Au, Bi, Co, V
Meropenem	532, 633, 785, 1064	Co, Cu
Ceftazidime	532, 633, 785, 1064	Ag [39]

## Data Availability

The original data presented in the study are openly available in Mendeley Data at https://data.mendeley.com/datasets/7r3b7svjny/1, accessed on 26 October 2024. DOI: 10.17632/7r3b7svjny.1.

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
