# Peer review of "Identifying the Molecular Fingerprint of Beta-Lactams via Raman/SERS Spectroscopy Using Unconventional Nanoparticles for Antimicrobial Stewardship"

_antibiotics, 2024, doi:10.3390/antibiotics13121157_

Round 1
Reviewer 1 Report
Comments and Suggestions for Authors
The manuscript is titled " Identifying Molecular Fingerprint of Beta-Lactams via Raman/SERS Spectroscopy using Unconventional Nanoparticles for Antimicrobial Stewardship" Researchers illustrate the feasibility of detecting spectra with characteristic vibrations (fingerprints) of various antimicrobials by Surface Enhanced Raman Spectroscopy. The authors' findings indicate novel methodologies for molecular drug monitoring through optical approaches utilizing unconventional nanoparticles. However, it is essential to offer further clarification on specific issues. Consequently, it is advised that this paper be published with just minor edits.
Table 6 wasn’t mentioned in the text and not explained.
In which form were Au, Ag, Co, Cu, Bi, and V purchased from Sigma? Please elaborate on nanoparticles synthesis section in detail.
Explain the mechanism of signal intensification via nanoparticles in detail? Does each nanoparticle have the same capability of sensing? If not, what is the reason behind this difference?
Haven’t you decorated nanoparticles with a certain functional group? Or use all of them as they are? Normally for these kinds of applications, surfaces are decorated/functionalized.
DLS results should also be presented. Even though it was mentioned that size distribution was studied, results are not included in the text. What is the size of nanoparticles? How would it affect the signal intensification? Would that be a reason for having different signal intensification capabilities?
ATB concentration and nano colloid concentration ratios are not the same. All 3 ATBs have tested at different concentrations with respect to the same colloid concentration which is 100mM. Why were ATB concentrations not chosen the same?
There is no TABLE 2!
Table 5 was mentioned after the Table. You should mention it before the TABLE.
In the Reference section, Reference 12 should be edited.
Author Response
We would like to thank Reviewer 1 for all the significant and important comments and suggestions.
Reviewer 1:
The manuscript is titled "Identifying Molecular Fingerprint of Beta-Lactams via Raman/SERS Spectroscopy using Unconventional Nanoparticles for Antimicrobial Stewardship" Researchers illustrate the feasibility of detecting spectra with characteristic vibrations (fingerprints) of various antimicrobials by Surface Enhanced Raman Spectroscopy. The authors' findings indicate novel methodologies for molecular drug monitoring through optical approaches utilizing unconventional nanoparticles. However, it is essential to offer further clarification on specific issues. Consequently, it is advised that this paper be published with just minor edits.
Comment 1: Table 6 wasn’t mentioned in the text and not explained.
Response 1: We are sorry for that. In fact, it should be written “Table 6”. We therefore added the sentence (page 17, first paragraph, line 532): Table 5 shows a summary of the obtained experimental results listing for each ATB the wavelengths for which the Raman spectra were recorded and for which NPs an enhanced Raman spectra (SERS) could be obtained.
Comment 2: In which form were Au, Ag, Co, Cu, Bi, and V purchased from Sigma? Please elaborate on nanoparticles synthesis section in detail.
Response 2: Since all nanoparticles were synthesized in our lab, we added the sentence “All nanoparticles used in this study were synthesized in our lab through LASIS (Laser Ablation Synthesis in Solution)” in the beginning of section 2.4 (Nanoparticle synthesis and characterization). In this same section, in line 247, we added the information: The metal target foils were purchased from Sigma-Aldrich with 99.99% purity.
Comment 3: Explain the mechanism of signal intensification via nanoparticles in detail? Does each nanoparticle have the same capability of sensing? If not, what is the reason behind this difference?
Response 3: We would like to thank the reviewer for this important consideration and decided to clarify this question by modifying the 9th paragraph (Introduction, line 101). The new paragraph version is now:
However, Raman spectroscopy has limitations related to fluorescence or low-intensity scattered signals when analyte concentration is small. Surface Enhanced Raman Spectroscopy (SERS) comes as a solution to these problems [18,19]. When a material (analyte) is adsorbed in the vicinity of nanostructures, interactions occur, such as charge transfer (a chemical effect) and also a significant increase in the electric field of the incident light (a physical effect called plasmonic resonance, more accentuated in the presence of metallic nanostructures) [16,18,25,26]. The chemical effect mechanism involves the interaction of adsorbed molecules with the metallic surface (chemisorption), resulting in polarizability changes or metal-adsorbate charge-transfer, leading to the creation of new molecular states due to this direct metal-analyte interaction, and peak shifts in SERS spectra as compared to Raman. On the other hand, the plasmonic electric field enhancement allows for the occurrence of SERS even in circumstances in which analytes are not directly adsorbed on the metal but are within a few nanometers from it (in places called hotspots). Currently, most authors agree that the chemical mechanism provides enhancements of only a few orders of magnitude whereas the plasmonic enhancement mechanism is the dominant contribution to SERS [16] and can imply amplification of the Raman signal by factors of up to 1012, thus enabling even the detection of a single molecule [27,28]. In this sense, exploring specific interactions between different combinations of analytes and nanomaterials (nanoparticles and nanostructured surfaces) is an open field of investigation, for the determination of sensing capabilities in diversified configurations.
Comment 4: Haven’t you decorated nanoparticles with a certain functional group? Or use all of them as they are? Normally for these kinds of applications, surfaces are decorated/functionalized.
Response 4: We thank the reviewer for this question which allowed us to better elaborate on the fact that our experiments are indeed label-free. Our approach does not require surface functionalization of nanoparticles, and our results relate to an ongoing trend in SERS applications (label-free); therefore, we included a complement in section 4 (Discussion), line 558, and added three new references to our text:
In addition, our approach does not require nanoparticle functionalization and can therefore be characterized as a label-free method [46-49].
Comment 5: DLS results should also be presented. Even though it was mentioned that size distribution was studied, results are not included in the text. What is the size of nanoparticles? How would it affect the signal intensification? Would that be a reason for having different signal intensification capabilities?
Response 5: Once again, we appreciate the reviewer's comment. We decided to modify Figure 7 (now split in 7a and 7b). In the new version of our manuscript. Figure 7 now incorporates DLS results as suggested by the referee. In Figure 7a we present UV-Vis graphs (as we did in the previous version), and in Figure 7b, the corresponding DLS images include colloidal size distribution information. We therefore modified our text in section 3 (Results), updated the figure's legend, and rewrote the paragraph starting at line 389:
The nanoparticle colloids were prepared using the previously reported procedure, where the target of each metal was immersed in pure water and irradiated by the laser. The optical characterization of the suspensions was performed by UV-Vis spectroscopy, and through DLS we obtained the size distribution. In Figure 7a, we present the optical extinction of all colloidal nanoparticles. Noticeably, these colloids exhibit plasmon bands that are characteristic of spherical nanoparticles; whilst gold and copper present peaks in the visible region (around 520 nm and 600 nm, respectively), the other elements present peaks in the ultraviolet region. Figure 7b summarizes DLS measurements for all nanoparticles. DLS typically provides larger diameter values compared to those obtained from more direct measurement techniques; the discrepancy arising because DLS measures the diffusion behavior of particles in solution, accounting not only for the particle's core size but also for the surrounding solvation layer and any particle-particle interactions. Therefore, DLS tends to overestimate particle size, and the values for average size and dispersion measured are biased to larger particles, yet this technique allows us to compare size distribution of nanoparticles from the different materials used in our experiments. It is worth to mention that the broader size distribution observed in Figure 7b arises from the intrinsic nature of the LASiS process, where rapid and localized energy deposition on the target material gives rise to particles of varying sizes within a single ablation event [35]. In addition, factors such as the laser parameters, pulse duration, and the dynamics of bubble formation and collapse in the liquid environment depend on the specific metal target used in the synthesis, which explains the heterogeneity in nanoparticle size observed for the different colloidal solutions in Figure 7b.
Regarding the question on how nanoparticle size would affect the signal intensification, and whether it would be a reason for having different signal intensification capabilities, we understand it should be a matter for further investigation, and we intend on designing new experiments in which these questions would be addressed.
Comment 6: ATB concentration and nano colloid concentration ratios are not the same. All 3 ATBs have tested at different concentrations with respect to the same colloid concentration which is 100mM. Why were ATB concentrations not chosen the same?
Response 6: We would like to thank the referee since Comment 6 allowed us to rewrite and improve section 2.2, regarding sample preparation. We modified our text starting at line 187:
The samples were prepared using borosilicate microscopy glass slides (Knittelglass, Germany) as substrates. In the Raman experiments, pure solid samples in powder form were used for measuring the Raman spectra of each selected ATB. Powder samples were also diluted in water (deionized, conductivity less than 5 mS) at concentrations of 100 mM, thus a 1 μL drop was deposited on the glass slide and left to dry. These dried drops were used as a reference since for each ATB of choice a clear Raman signal could be obtained and compared with the corresponding powder Raman spectra.
The ATB solutions were then further diluted in pure water several times and the dried drops were tested over the slides without nanoparticles until Raman signals could not be detected (under the same experimental configurations used for the 100 mM concentration drops). Once no Raman signal could be measured, we used those over diluted samples as a starting sample concentration for verifying SERS occurrence. For the SERS substrates preparation, a 1 μL drop of each nanoparticle colloid (concentration of 0.01 mg/ml) was deposited on the glass slide and let to dry for film formation, therefore constituting our SERS substrates upon which ATB drops were deposited. On the special case of CoNP which is a paramagnetic material, the slide was placed over a neodymium magnet (ca. 270 mT) to obtain a more homogeneous substrate. Therefore, for the SERS experiments, ATB concentrations that depended upon the specific nanoparticle-ATB combination were used.
Comment 7: There is no TABLE 2!
Response 7: We are sorry for the missing table index. In the new version of our manuscript, we corrected all table numbers and their corresponding captions.
Comment 8: Table 5 was mentioned after the Table. You should mention it before the TABLE.
Response 8: We agree with the referee and added a sentence to the end of the second paragraph (line 365): The main vibrational bands are listed in Table 4, with their respective attributions based on comparisons with data already published in the literature.
Comment 9: In the Reference section, Reference 12 should be edited.
Response 9: Reference 12 was corrected to adhere to the citation standard:
Marangoni, C.G.P. da S.; Machado, T.N.; Thaler, J.; dos Anjos, V.P.; Barros, F.S.; Costa, L.M.D.; de Góes, R.E.; Schreiner, W.H.; Bezerra Jr, A.G. Detecção e Caracterização de Antimicrobianos Usando Espectroscopia Raman Amplificada por Superfície. Braz J Infect Dis 2022, 26, doi:10.1016/j.bjid.2021.101999.
Reviewer 2 Report
Comments and Suggestions for Authors
The manuscript “Identifying Molecular Fingerprint of Beta-Lactams via Raman/SERS Spectroscopy using Unconventional Nanoparticles for Antimicrobial Stewardship” presents an interesting study on the detection of three beta-lactam antibiotics—ampicillin (penicillin subclass), meropenem (carbapenem subclass), and ceftazidime (cephalosporin subclass)—using Surface-Enhanced Raman Spectroscopy (SERS) with various metallic nanoparticles.
This study is well-executed, and the findings provide valuable insights into the field of antibiotic detection, offering potential contributions to antimicrobial resistance research. However, several revisions, particularly in the introduction, are recommended to improve clarity and scientific rigor:
- “with this several studies from academia and the pharmaceutical industry have tried to identify methods to optimize the use of antimicrobials to combat resistant pathogens” sounds strange and should be reformulated.
- “In line with this, we have Raman spectroscopy,” should also be rewritten
- “This is based on an analytical technique called Raman spectroscopy, which consists of illuminating the material to be analyzed and detecting the scattered light, which contains specific information about the molecular constituents that make up the material for part of the incident energy excites molecular vibrations.” is overly simplistic and should be expanded to more accurately describe the process and importance of Raman scattering.
- “and also a significant increase in the electric field (…) of the incident light” is misleading, as the increase in the electric field occurs in the localized area where the incident light interacts with the nanoparticle, rather than in the incident light itself. This sentence should be revised to reflect this nuance.
- “(because each sample interrogation…” also sounds strange and should be rewritten.
- “The equipment operated in the 785 nm (infrared) and 1064 nm (infrared) bands, with power ranging from 0 mW to 450 mW.” Please replace “bands” with “wavelengths,”. Also, “0 mW” power should be clarified, as this value does not seem appropriate for the context.
Once these modifications have been addressed, I believe the manuscript will be well-suited for consideration for publication in Antibiotics.
Author Response
We would like to thank Reviewer 2 for all the important comments and suggestions.
REVIEWER 2:
The manuscript “Identifying Molecular Fingerprint of Beta-Lactams via Raman/SERS Spectroscopy using Unconventional Nanoparticles for Antimicrobial Stewardship” presents an interesting study on the detection of three beta-lactam antibiotics—ampicillin (penicillin subclass), meropenem (carbapenem subclass), and ceftazidime (cephalosporin subclass)—using Surface-Enhanced Raman Spectroscopy (SERS) with various metallic nanoparticles.
This study is well-executed, and the findings provide valuable insights into the field of antibiotic detection, offering potential contributions to antimicrobial resistance research. However, several revisions, particularly in the introduction, are recommended to improve clarity and scientific rigor:
Comment 1: “with this several studies from academia and the pharmaceutical industry have tried to identify methods to optimize the use of antimicrobials to combat resistant pathogens” sounds strange and should be reformulated.
Response 1: we changed the text on line 65 to:
The Antimicrobial Stewardship program has been worked on since the 1990s, however, in recent years, with the incorporation of One Health (animal, plant and environmental health), the term has become popular, bringing an even more multidisciplinary vision [14]. Given that, several studies from academia and the pharmaceutical industry have tried to identify methods to optimize the use of antimicrobials to combat resistant pathogens [2].
Comment 2: “In line with this, we have Raman spectroscopy,” should also be rewritten
Response 2: we changed the text on line 82 to:
“In line with these necessities, Raman spectroscopy has several applications in the health area, such as polymorphic study, identification of raw materials, identification of "counterfeiting", determination of the quantity as well as homogeneity determination of active pharmaceutical ingredients (API) [18–20].
Comment 3: “This is based on an analytical technique called Raman spectroscopy, which consists of illuminating the material to be analyzed and detecting the scattered light, which contains specific information about the molecular constituents that make up the material for part of the incident energy excites molecular vibrations.” is overly simplistic and should be expanded to more accurately describe the process and importance of Raman scattering.
Response 3: We changed the text starting on line 85, extending the description and highlighting its importance:
This technique involves illuminating a material and detecting the scattered light, which carries distinct spectral information. Part of the incident energy interacts with the molecular vibrations of the material, resulting in characteristic Raman peaks that serve as a molecular fingerprint. These peaks enable the identification of chemical bonds and functional groups present in the sample, making Raman spectroscopy a valuable photonic tool for both qualitative and quantitative analysis. The set of these vibrations, which appear in the form of peaks in the spectrum of the scattered light, forms a fingerprint of the material [21,22]. Furthermore, its versatility allows non-destructive analysis of solid, liquid, and gaseous samples, with applications extending over fields such as materials science, environmental monitoring, and pharmaceuticals and biomedicine. In healthcare, Raman spectroscopy holds transformative potential, enabling non-invasive measurement of drug concentrations in a patient’s bloodstream, minimizing the need for extensive blood collection or other invasive procedures [16]. Additionally, it facilitates the identification of protein-based drug substances in a non-destructive manner [23,24], reinforcing its critical approach in advancing precision medicine and pharmaceutical quality control.
Comment 4: “and also a significant increase in the electric field (…) of the incident light” is misleading, as the increase in the electric field occurs in the localized area where the incident light interacts with the nanoparticle, rather than in the incident light itself. This sentence should be revised to reflect this nuance.
Response 4: This explanation was addressed by Comment 3 for reviewer 1.
Comment 5: “(because each sample interrogation…” also sounds strange and should be rewritten.
Response 5: we have rewritten the text to (Section 1, line 122):
The usual laboratory methods for drug monitoring were compared with the SERS technique in Table 1. This comparison highlights that with further development and refinement for each specific sample, it may become a powerful tool for drug monitoring in the future. This approach also depends on parameters such as laser power, exposure time, and type of acquisition - point, map, or image [29].
Comment 6: “The equipment operated in the 785 nm (infrared) and 1064 nm (infrared) bands, with power ranging from 0 mW to 450 mW.” Please replace “bands” with “wavelengths,”. Also, “0 mW” power should be clarified, as this value does not seem appropriate for the context.
Response 6: We agree with the referee, and made the suggested modifications (see Section 2.3, line 211):
The equipment operated at 785 nm and 1064 nm wavelengths with power ranging from 2 µW to 450 mW.
Comment 7: Once these modifications have been addressed, I believe the manuscript will be well-suited for consideration for publication in Antibiotics.
Response 7: We thank the reviwer for the important insights on our work. After the suggested modifications the text has been significantly improved.
Reviewer 3 Report
Comments and Suggestions for Authors
The manuscript by Pereira dos Anjos et al. is about the detection of antibiotics of the beta-lactam family by Raman and SERS spectroscopies using unconventional nanoparticles. The aim is to develop methods using nanoparticles to detect by SERS the antibiotics and eventually, monitor these antibiotics in biological settings (ex blood sample). The study is well constructed and appropriately carried out. Raman spectra of beta-lactams antibiotics included in the manuscript were reported before in the litterature and are essentially control experiments in the current manuscript. The new findings are essentially about the nature of the nanoparticles that were used for SERS. Instead of the usual Au nanoparticles, the authors used nanoparticles made of Bi, Co, Cu and V to enhance the Raman signal. Great signal enhancement were indeed observed. However, it is unclear to me whether SERS would really be a valuable analytical tool in real biological settings where hundreds of compounds different from the antibiotic would be present. There is little discussion about the improvement afforded by using nanoparticles made of Au, Bi Co, Cu and V instead of Au, besides the price.
Specific comments
The quality of English should be improved through the manuscript
In all Figures, the wavenumber of the main peaks should be identified directly on the Figures.
Line 79 … a the molecular scale have significantly…
Line 87 … of the incident energy that excites…
Line 94 However, Raman spectroscopy has limitations…
Line 116 Modification of the antibiotic is only one of possible mechanism of resistance to antitiotics.
Line170 … for the sample preparation follows, a drop with 1….
Line 171…glass slide and let to dry..
Table 3, 4 and 5. Indicate if the assignment is for a stretching, bending or breathing modes.
Line 308 The last two 84 vibration peaks are related to the vibrations of the beta-lactam ring. This sentence does not make sense.
Lines 314-315 It is not evident from the spectra presented that the resolution is better with an excitation wavelength of 785 nm compared to 1064 nm.
Line 358 …has been found for its assignment.
Line 369 Define DLS
Line 377 I do not understand the sentence ‘During the experiment with ampicillin, several measurements were made, and consequently, the spectrum was not presented most of the time’.
Line 386 …indeed a SERS measurement.
Figure 8 The spectra in the powder form and by SERS look very different. What are the peaks that are in common in these spectra?
Line 395 and 396 Explain why the bands are shifted and provide a reference for the explanation. I thought that SERS only enhanced the signal of some Raman bands, not that it caused spectral shifts.
There is no Figure 11, so why is there a Figure 12?
Line 463 The purpose of this experiment…
Comments on the Quality of English Language
Several sentences are very long and many are not well constructed.
Author Response
We would like to thank Reviewer 3 for all the important comments and suggestions.
Reviewer 3:
General comment: The manuscript by Pereira dos Anjos et al. is about the detection of antibiotics of the beta-lactam family by Raman and SERS spectroscopies using unconventional nanoparticles. The aim is to develop methods using nanoparticles to detect by SERS the antibiotics and eventually, monitor these antibiotics in biological settings (ex blood sample). The study is well constructed and appropriately carried out. Raman spectra of beta-lactams antibiotics included in the manuscript were reported before in the litterature and are essentially control experiments in the current manuscript. The new findings are essentially about the nature of the nanoparticles that were used for SERS. Instead of the usual Au nanoparticles, the authors used nanoparticles made of Bi, Co, Cu and V to enhance the Raman signal. Great signal enhancement were indeed observed. However, it is unclear to me whether SERS would really be a valuable analytical tool in real biological settings where hundreds of compounds different from the antibiotic would be present. There is little discussion about the improvement afforded by using nanoparticles made of Au, Bi Co, Cu and V instead of Au, besides the price.
Response to general comment: In section 3 (Results), we rewrote several paragraphs (see, for instance, our responses to Reviewer's Comments 14 and 16). In addition, we added new information in section 4 (Discussion) on the importance of the “label-free” nature of the experiments performed with our unconventional nanoparticles (in response to Reviewer 1, Comment 4). Also, in the discussion (line 564), we added: ... not only reducing costs, but also acting as SERS agents in experimental setups where noble metal proved not to be efficient substrate materials, as shown in the experiments we performed with ampicillin.
We now believe that we gave a satisfactory answer to the general comment of Reviwer 3.
Specific comments:
Comment 1: The quality of English should be improved through the manuscript.
Response 1: Recently, our university provided a language reviewing tool (Grammarly) available to faculty and grad students. Therefore, we utilized Grammarly to review possible English mistakes and enhance the clarity and quality of the text in our paper.
Comment 2: In all Figures, the wavenumber of the main peaks should be identified directly on the Figures.
Response 2: Given that the band assignments are tentative and obtained from comparison with other works in the literature, we opted to maintain both measured and theoretical band assignments in the tables (see for example, Tables 2, 3 and 4). Specific work on simulating, measuring and assigning these peaks may be covered in future work. In addition, as stated in our abstract, “Our experiments demonstrate the possibility of identifying spectra with characteristic vibrations (fingerprints) ... via SERS. Our results point to new strategies for molecular monitoring of drugs by optical techniques using unconventional nanoparticles.”
Comment 3: Line 79 “… a the molecular scale have significantly…”
Response 3: the text was changed to: In this context, studies in the field of Nanotechnology have been promising in health basic and translational research, as the ability to manipulate, monitor, and identify structures at the molecular scale has significantly impacted the search for diagnostic methods and drug administration [17].
Comment 4: Line 87 … of the incident energy that excites…
Response 4: We made changes to the text according to the suggestions of Reviwer 2 (see Response to Reviewer's Comment 3).
Comment 5: Line 94 However, Raman spectroscopy has limitations…
Response 5: We made changes to address this issue (see line 101, in the new version): However, Raman spectroscopy has limitations related to...
Comment 6: Line 116 Modification of the antibiotic is only one of possible mechanism of resistance to antibiotics.
Response 6: We agree with the reviwer. Indeed, modification of the antibiotic is one of the (many) possible mechanisms of resistance, and there are certainly other mechanisms, however, our technique aims to monitor peaks alterations which, in turn, indicate changes in the molecular structure.
Comment 7: Line 170 … for the sample preparation follows, a drop with 1….
Response 7: We made changes to the text according to the suggestions of Reviwer 2 (see Response to Reviewer's Comment 3).
Comment 8: Line 171…glass slide and let to dry..
Response 8: see line 200 in the new version: deposited on the glass slide and left to dry
Comment 9: Table 3, 4 and 5. Indicate if the assignment is for a stretching, bending or breathing modes.
Response 9: We understand this comment was already addressed in our previous response (see Comment 2, Reviwer 2).
Comment 10: Line 308 The last two 84 vibration peaks are related to the vibrations of the beta-lactam ring. This sentence does not make sense.
Response 10: There had been a typo, and we regret that. Modifications were made (see line 329 in the new version): The last two vibration peaks...
Comment 11: Lines 314-315 It is not evident from the spectra presented that the resolution is better with an excitation wavelength of 785 nm compared to 1064 nm.
Response 11: The change in resolution is more apparent when comparing 532 nm with the larger wavelengths. The text was changed to reflect that (see line 335): The same is expected to occur in IR, although to a lesser degree (Anton Paar Raman spectrometer).
Comment 12: Line 358 …has been found for its assignment.
Response 12: We made a correction (see line 379 in the new version): ...literature, since little research material has been found for its band assignment.
Comment 13: Line 369 Define DLS
Response 13: It has alrealdy been defined in line 266 (dynamic light scattering).
Comment 14: Line 377 I do not understand the sentence ‘During the experiment with ampicillin, several measurements were made, and consequently, the spectrum was not presented most of the time’.
Response 14: We agree and would like to thank the referee for the comment. Therefore, we decided to write a new version of the paragraph (see line 421 in the new manuscript version):
An interesting feature in the experiments performed with ampicillin illustrated the importance of using nonconventional nanoparticles as SERS substrates alternative to noble metals. Although some experiments were performed with silver nanoparticles, it always led to local sample burning under the experimental conditions available for the SERS measurement setup. Similarly, when gold nanoparticles were used as substrates, no signal amplification could usually be observed since the integration time selected had to be noticeably short (to prevent sample burning). After several measurements, only a few SERS signals were detected with AuNP substrates, for which Figure 8 shows a typical spectrum.
Comment 15: Line 386 …indeed a SERS measureme
Response 15: We corrected in the text (see line 434 in the new version) “...which leads us to conclude that this is indeed a SERS measurement.
Comment 16: Figure 8 The spectra in the powder form and by SERS look very different. What are the peaks that are in common in these spectra?
Response 16: We thank the reviwer for another important comment. We opted to rewrite parts of the paragraph (starting at line 433):
The first relevant result measurements performed with AuNP is that even for a very short integration time the characteristic spectrum of ampicillin was obtained, which leads us to conclude that this is indeed a SERS measurement. Another interesting fact is that the two peaks related to the beta-lactam ring (1692 and 1782 cm−1, on the lower right corner of Figure 8) were not evident in this measurement. Something that may justify this is that this measurement, unlike the other measurements reported here, was not obtained through a map that scanned an area of the sample, but with exploratory point measurements in different regions of the sample. Perhaps if the integration time were longer, both peaks would be resolved; however, with a longer integration time, the sample would be burned locally. It is important to emphasize that biological samples and the drugs used in this research are fragile, so parameters such as excitation laser power, and even NPs used can contribute to the sample being degraded quickly at the time of interrogation, which probably explains the large shifts observed in Figure 8. The most intense band was shifted by +94 cm−1, the 1201 cm−1 band was displaced by +164 cm−1, and the vibration at 1600 cm−1 was shifted to 1613 cm−1. In summary, despite its widespread use as SERS agents, gold nanoparticles did not perform satisfactorily and proved not to be efficient substrate materials in these experiments with ampicillin.
Comment 17: Line 395 and 396 Explain why the bands are shifted and provide a reference for the explanation. I thought that SERS only enhanced the signal of some Raman bands, not that it caused spectral shifts.
Response 17:
Although we understand that it is an important issue, we believe that it is not the aim of our article to go through this discussion, especially since it is an open question in literature. We wrote a short paragraph that could be added to our manuscript, but we believed this would make the text too long and unnecessarily wordy (therefore, we prefer not to include it in our final version):
The affinity of analytes for different nanoparticle surfaces can result in varying spectral shifts in spectral peaks when compared to conventional Raman spectroscopy due to several underlying mechanisms. These include charge transfer processes between the analyte and the metallic surface, and the Stark effect, where local electric fields influence vibrational frequencies. The binding interactions and changes in molecular symmetry caused by adsorption onto the metallic surface also play a significant role; in addition, certain specific interactions with biological molecules can further modulate these shifts. Therefore, the precise factors governing these frequency shifts remain an open question. Further information could be found in:
Ma, H., Liu, S., Zheng, N., Liu, Y., Han, X. X., He, C., Lu, H., & Zhao, B. (2019). Frequency Shifts in Surface-Enhanced Raman Spectroscopy-Based Immunoassays: Mechanistic Insights and Application in Protein Carbonylation Detection. Analytical Chemistry, 91, 9376–9381. DOI: 10.1021/acs.analchem.9b02640
Comment 18: There is no Figure 11, so why is there a Figure 12?
Response 18: We are sorry for the missing figure 11. We corrected all the captions and its reference in the text.
Comment 19: Line 463 The purpose of this experiment…
Response 19: We rewrote the text (line 520). The original reviewed text was “The result of this experiment was to show that the nanoparticles contributed”, and we changed it to:
The purpose of this experiment was to show that the nanoparticles contributed to the signal intensification, and in the case where there was no NP, the characteristic spectrum was not obtained.